# Direct evidence for charge stripes in a layered cobalt oxide

P. Babkevich[1], P.G. Freeman[2], M. Enderle[3], D. Prabhakaran[4] & A.T. Boothroyd[4]

Recent experiments indicate that static stripe-like charge order is generic to the hole-doped copper oxide superconductors and competes with superconductivity. Here we show that a similar type of charge order is present in $La_{5/3}Sr_{1/3}CoO_4$, an insulating analogue of the copper oxide superconductors containing cobalt in place of copper. The stripe phase we have detected is accompanied by short-range, quasi-one-dimensional, antiferromagnetic order, and provides a natural explanation for the distinctive hourglass shape of the magnetic spectrum previously observed in neutron-scattering measurements of $La_{2-x}Sr_xCoO_4$ and many hole-doped copper oxide superconductors. The results establish a solid empirical basis for theories of the hourglass spectrum built on short-range, quasi-static, stripe correlations.

[1] Laboratory for Quantum Magnetism, Institute of Physics, École Polytechnique Fédérale de Lausanne, CH-1015 Lausanne, Switzerland. [2] Jeremiah Horrocks Institute for Mathematics, Physics and Astronomy, University of Central Lancashire, Preston, Lancashire PR1 2HE, UK. [3] Institut Laue-Langevin, CS 20156, 38042 Grenoble Cedex 9, France. [4] Department of Physics, University of Oxford, Clarendon Laboratory, Oxford OX1 3PU, UK. Correspondence and requests for materials should be addressed to P.B. (email: peter.babkevich@epfl.ch) or to A.T.B. (email: a.boothroyd@physics.ox.ac.uk).

The hourglass spectrum has been observed in measurements on $La_{2-x}Sr_xCoO_4$ with $x = 0.25$–$0.4$ (refs 1–3); however, its origin has become a matter of contention. Initially[1], it was explained by a model of short-range spin and charge stripe correlations similar to those found in certain hole-doped copper-oxide superconductors near one-eighth doping[4]. However, the subsequent failure to observe charge stripe order in diffraction experiments led Drees et al. to propose a stripe-free model based on a nanoscopically phase-separated ground state comprising coexisting regions with local composition $x = 0$ and $x = 0.5$ (ref. 3). The presence or absence of charge stripes, therefore, is of crucial importance for understanding the hourglass spectrum in $La_{2-x}Sr_xCoO_4$.

The crystal structure of $La_{2-x}Sr_xCoO_4$ contains well-separated $CoO_2$ layers with Co arranged on a square lattice. These layers are isostructural with the $CuO_2$ layers in the copper oxide high-temperature superconductors. The parent compound $La_2CoO_4$ is an insulator with Néel-type antiferromagnetic (AFM) order below $T_N = 275$ K (ref. 5). Substitution of Sr for La adds positive charge (holes) on the $CoO_2$ layers, converting $Co^{2+}$ into $Co^{3+}$ and suppressing the AFM order. In the compositions of interest here the $Co^{2+}$ ions have spin $S = 3/2$ and the $Co^{3+}$ ions carry no spin ($S = 0$; refs 1,6).

$La_{2-x}Sr_xCoO_4$ compounds with $x \gtrsim 0.25$ exhibit diagonally modulated incommensurate AFM order with a modulation period that varies linearly with doping from $x \simeq 0.25$ up to at least $x = 0.5$ (refs 2,7). Such magnetic behaviour is also found in the isostructural layered nickelates $La_{2-x}Sr_xNiO_4$, where it is caused by ordering of the holes into stripes that align at 45° to the Ni-O bonds and that act as antiphase domain walls in the AFM order[8–11]. The superstructure of spin and charge order observed in the nickelate composition of most relevance to the present work ($x = 1/3$) has three times the period of the square lattice. For $La_{2-x}Sr_xNiO_4$ with $x > 1/3$, the charge stripe period decreases linearly with increasing $x$ until at $x = 0.5$ the charge order approaches an ideal checkerboard pattern, see Fig. 1.

The linear doping dependence of the incommensurate magnetism observed in $La_{2-x}Sr_xCoO_4$ for $x < 1/2$ was interpreted[7] as possible evidence for the existence of diagonal spin-charge stripe phases similar to those found in $La_{2-x}Sr_xNiO_4$. Consistent with this, indirect evidence for charge freezing at $\sim 100$ K was inferred from magnetic resonance experiments on $La_{2-x}Sr_xCoO_4$ (refs 12,13). Until now, however, attempts to detect charge stripe order in the cobaltates more directly by neutron and X-ray diffraction have been unsuccessful. In fact, instead of the stripe phase expected by analogy with $La_{2-x}Sr_xNiO_4$, recent diffraction measurements[14] made on $La_{5/3}Sr_{1/3}CoO_4$ detected the presence of short-range checkerboard charge order, a very stable phase previously found in $La_{2-x}Sr_xCoO_4$ at $x = 1/2$ (refs 15,16).

In this work we used polarized neutron diffraction to detect spin and charge stripes in $La_{5/3}Sr_{1/3}CoO_4$. The stripes have the same arrangement of spin and charge order (SO and CO) as those found in $La_{2-x}Sr_xNiO_4$ with $x \simeq 1/3$ but have a higher degree of disorder. The presence of this stripe phase provides a natural explanation for the hourglass magnetic spectrum of $La_{5/3}Sr_{1/3}CoO_4$.

## Results

**Signatures of checkerboard and stripe correlations.** Neutrons do not couple directly to charge but can probe CO through the associated structural distortions. Polarization analysis was employed in this work to achieve an unambiguous separation of the structural and magnetic scattering from $La_{5/3}Sr_{1/3}CoO_4$ due to charge and spin order (see Methods). The CO and SO phases of

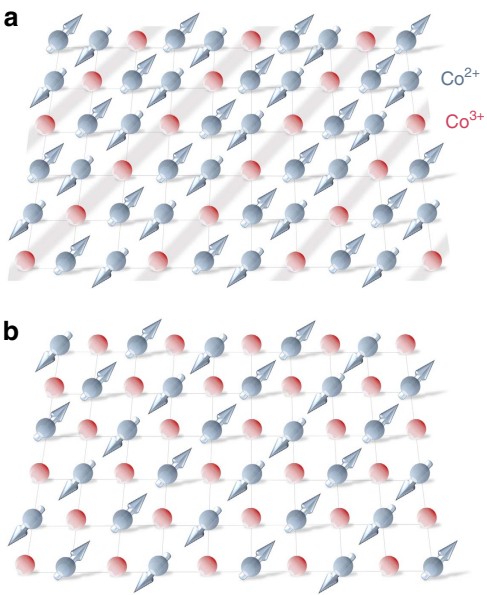

**Figure 1 | Models of ideal spin and charge order in $La_{2-x}Sr_xCoO_4$.** (**a**) Period-3 spin and charge stripe order for $x = 1/3$. (**b**) Checkerboard spin and charge order for $x = 1/2$. The arrows represent magnetic moments on $Co^{2+}$ ions and the red spheres denote $Co^{3+}$. The ions are located at the corners of the square lattice motif. The same patterns of spin and charge order apply to $La_{2-x}Sr_xNiO_4$.

most relevance to our diffraction results are depicted in Fig. 1. These are the period-3 stripe phase and the checkerboard phase.

Figure 2a is a sketch of the $(h, h, l)$ plane in reciprocal space showing the main scattering features investigated here. The reciprocal lattice is defined with respect to the conventional tetragonal unit cell (space group I4/mmm, lattice parameters $a = 0.386$ nm, $c = 1.26$ nm), which describes the room-temperature crystal structure of $La_{5/3}Sr_{1/3}CoO_4$. Diffraction scans along the paths marked A, B and C are presented in Fig. 2b–d from measurements made at a temperature $T = 2$ K. Scan A in Fig. 2b shows two broad magnetic peaks centred close to $h = 1/3$ and $2/3$, respectively. These derive from the diagonally modulated AFM short-range order observed previously[1,7], which produces a fourfold pattern of magnetic diffraction peaks centred on the Néel AFM wavevectors $\mathbf{Q}_{AF} = (h + 1/2, k + 1/2, l)$ with integer $h$, $k$, $l$. The magnetic peaks are displaced from $\mathbf{Q}_{AF}$ by $\pm (\epsilon_{so}/2, \epsilon_{so}/2, 0)$ and $\pm (\epsilon_{so}/2, -\epsilon_{so}/2, 0)$, where $\epsilon_{so} \approx x$ for doping levels between 0.25 and 0.5 (refs 2,7). From the fitted magnetic peak positions we find $\epsilon_{so} = 0.37 \pm 0.01$, and from their widths (half width at half maximum) we determine a magnetic correlation length of $\xi_{so} = 0.68 \pm 0.06$ nm in the [110] direction, consistent with the previous unpolarized neutron diffraction study[1].

The non-magnetic signal in scan A contains a peak at $h = 0.5$, with two small shoulders on its flanks at about the same positions as the magnetic peaks (Fig. 2b). Figure 3a compares the same scan at temperatures of 2 and 300 K. The shoulders of scattering near $h = 1/3$ and $2/3$ are no longer present at 300 K, whereas the amplitude of the peak at $h = 0.5$ remains virtually unchanged. The difference between the 2 and 300 K data sets, also shown in Fig. 3b, reveals two peaks centred approximately on $h = 1/3$ and $2/3$. The presence of this additional structural scattering at low temperatures was confirmed in several other scans parallel to scan A, two of which are shown in Fig. 3c–f. These reproduce the same key features as observed in Fig. 3a,b: the central peak does not

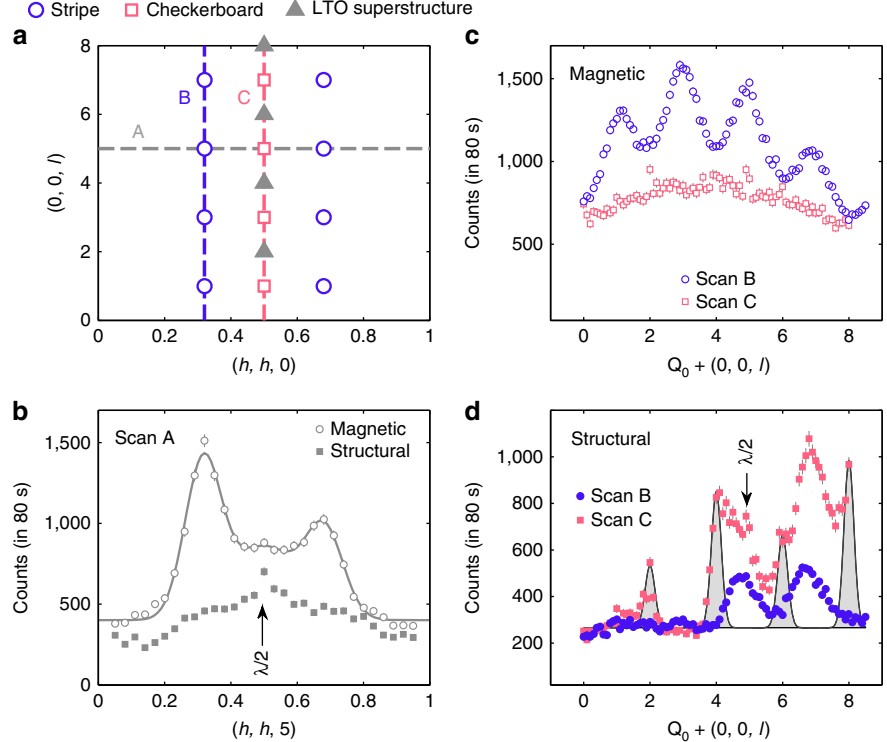

**Figure 2 | Neutron diffraction from La$_{5/3}$Sr$_{1/3}$CoO$_4$ at 2 K.** (**a**) Diagram of the ($h$, $h$, $l$) reciprocal lattice plane showing positions of stripe, checkerboard CO and LTO superstructure peaks. (**b**) Scans along path A showing magnetic and structural signals separated by polarization analysis. (**c**,**d**) Magnetic and structural scattering measured in scans along paths B and C. In **b**,**d**, small spurious peaks from half wavelength ($\lambda/2$) contamination in the neutron beam are indicated. The shaded peaks in **d** indicate low-temperature orthorhombic (LTO) superlattice reflections.

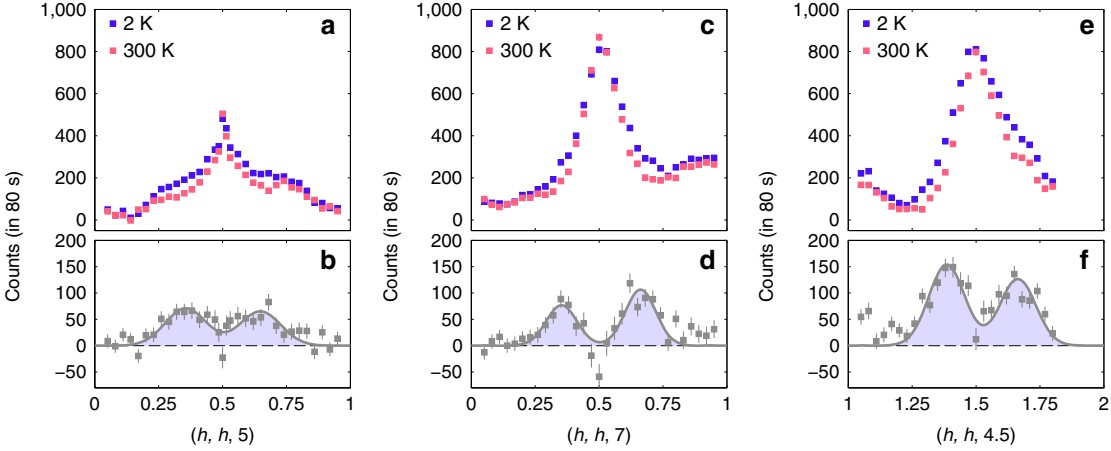

**Figure 3 | Charge order scattering from La$_{5/3}$Sr$_{1/3}$CoO$_4$.** Structural scattering intensities are compared at temperatures of 2 and 300 K in scans along (**a**,**b**) ($h$, $h$, 5), (**c**,**d**) ($h$, $h$, 7) and (**e**,**f**) ($h$, $h$, 4.5). The spurious peaks at $h \simeq 0.25$ and 0.75 in **a**, at $h \simeq 0.9$ in **c** and at $h \simeq 1.1$ and 1.7 in **e** are aluminium powder peaks. The lower plots in each panel show the difference between the curves recorded at 2 and 300 K. The shaded peaks are fits of two Gaussian peaks to these data.

change in amplitude between 2 and 300 K, but additional diffuse scattering peaks centred at $h \simeq 1/3$ and 2/3 develop below 300 K.

This excess structural diffuse scattering at low temperature is consistent with the presence of charge stripes. In the stripe model the diagonal charge modulation gives rise to CO diffraction peaks, which are displaced from the reciprocal lattice points by $\pm(\epsilon_{co}, \epsilon_{co}, 0)$, where $\epsilon_{co} = \epsilon_{so}$. From our data we obtain $\epsilon_{co} = 0.36 \pm 0.01$, in agreement with $\epsilon_{so} = 0.37 \pm 0.01$ found earlier. The fact that $\epsilon_{so}$ and $\epsilon_{co}$ are greater than 1/3, the value for ideal period-3 stripes (Fig. 1a), suggests either that the hole-doping level of our crystal could be slightly in excess of 1/3, or that the

presence of defects in an otherwise ideal period-3 stripe pattern causes an effective small incommensurability[17].

Because the stripes can run along either diagonal of the square lattice with equal probability, we also expect CO peaks at $\pm(\epsilon_{co}, -\epsilon_{co}, 0)$ relative to the reciprocal lattice points. This pair of peaks together with those at $\pm(\epsilon_{co}, \epsilon_{co}, 0)$ generate the characteristic fourfold pattern of CO peaks depicted in Fig. 4a that accompanies the fourfold pattern of SO peaks. To demonstrate this tetragonal anisotropy, we show in Fig. 4b,c the magnetic and structural scattering intensities measured along two arcs of the circular path indicated on Fig. 4a. We performed this

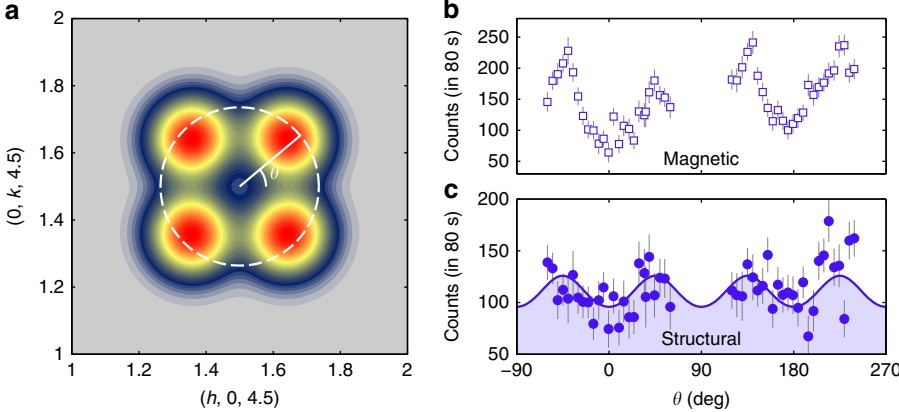

**Figure 4 | In-plane spin and charge scattering from stripes in La$_{5/3}$Sr$_{1/3}$CoO$_4$.** (**a**) Colour map of simulated charge stripe scattering intensity. (**b**) Magnetic and (**c**) structural scattering around the circular path shown in **a**. The intensity is the difference between measurements at 2 and 300 K. The line and shading in **c** are calculated from the fitted model shown in **a**.

type of scan by tilting the crystal to vary the wavevector component out of the $(h, h, l)$ scattering plane. The magnetic and charge scans both display the same periodic variation in intensity around the circular path, with the maxima located at the positions expected for stripe order. In Fig. 4a the charge peaks are depicted as isotropic (that is, circular), but diffuse scattering peaks from stripe order will in general display twofold symmetry as a result of different correlation lengths parallel and perpendicular to the stripes[17]. The SO peaks display this expected twofold anisotropy as reported previously[1]. However, more complete data would be needed to resolve any such anisotropy in the correlation lengths for the charge order.

Having established the presence of the stripe phase in La$_{5/3}$Sr$_{1/3}$CoO$_4$ we now develop a more complete picture of its complex ground state from the $l$ and temperature dependence of the diffuse scattering. Figure 2c presents scans of the magnetic diffuse scattering at 2 K along paths B and C. Scan B shows that the magnetic peaks are strongest at $l =$ odd integers in the $(h, h, l)$ plane. This is a consequence of the way the stripe order propagates along the $c$ axis (see Supplementary Note 1 and Supplementary Fig. 3). The large peak widths in scan B imply that the magnetic correlations do not extend much beyond an adjacent layer. The magnetic scattering at $\mathbf{Q}_{AF}$ (scan C) shows no periodic variation with $l$, consistent with a superposition of the four magnetic peaks that surround $\mathbf{Q}_{AF}$, one pair of which has maxima at $l =$ odd integers and the other at $l =$ even integers[1].

The $l$ variation of the structural diffuse scattering at 2 K shows a number of features of interest. Scan C (Fig. 2d) contains relatively sharp peaks at $l = 2, 4, 6$ and 8, with some diffuse scattering in between, which is strongest near $l = 5$ and 7. The sharp peaks at $l =$ even integers are superstructure reflections from the low-temperature orthorhombic (LTO) structural distortion that occurs in La$_{2-x}$Sr$_x$CoO$_4$ for $x \lesssim 0.4$ (refs 5,7). The intervening diffuse scattering has a similar variation with $l$ to that observed in the equivalent scan from the checkerboard CO phase in La$_{3/2}$Sr$_{1/2}$CoO$_4$ (refs 15,16,18). This confirms the presence of checkerboard CO in La$_{5/3}$Sr$_{1/3}$CoO$_4$, as reported previously[14], and accounts for the peaks at $h = 0.5$ and 1.5 in the scans shown in Figs 2b and 3. The observed $l$ dependence of this checkerboard CO scattering has been reproduced by a model for the pattern of subtle displacements of the in-plane and apical oxygen ions that surround the Co$^{2+}$ and Co$^{3+}$ sites[15,16,18].

Scan B (Fig. 2d), in which roughly half the intensity is from the stripe CO and half from the tail of the checkerboard CO peak, exhibits a similar $l$ dependence to that from the checkerboard CO

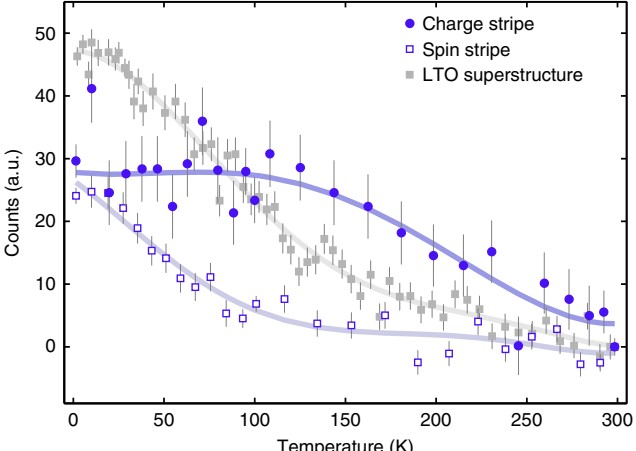

**Figure 5 | Temperature dependence of diffraction peaks.** The LTO superstructure data are from measurements at (0.5, 0.5, 4), and the spin and charge stripe data are from measurements at (1.38, 1.38, 4.5). In each case, a measurement made at 300 K has been subtracted from the remaining data. The lines are guides for the eye.

in scan C. Although the stripe CO and checkerboard CO peaks occur at different in-plane wave vectors, their $l$ dependence is expected to be similar because in both cases it originates from much the same local displacements of the oxygen ions in the CoO$_6$ octahedra. The structure factor of this local distortion mode is very small for $l \leq 3$ (refs 15,16,18), which may explain why previous attempts to detect stripe CO in La$_{5/3}$Sr$_{1/3}$CoO$_4$ from measurements at $l \leq 3$ have been unsuccessful.

**Temperature evolution of stripe spin and charge order.** Figure 5 shows the temperature dependence of the stripe and LTO superstructure peaks found in La$_{5/3}$Sr$_{1/3}$CoO$_4$. Charge stripe correlations develop continuously on cooling from room temperature down to 100 K, below which the signal remains constant. This temperature variation is consistent with the observation of charge freezing at $\sim 100$ K in magnetic resonance experiments[12,13]. The associated spin stripe-scattering begins to increase below $\sim 100$ K. The separation $\epsilon_{so}$ of the magnetic peaks changes only slightly between 2 and $\sim 90$ K (see Supplementary Note 2), consistent with the notion that the magnetic peaks are associated with the charge stripe order, which

is fully developed at 100 K. A similar behaviour was observed in $La_{5/3}Sr_{1/3}NiO_4$ (ref. 19). The LTO superstructure appears gradually below room temperature but develops most strongly below ~150 K, in accord with the published phase diagram of $La_{2-x}Sr_xCoO_4$ (ref. 7) and previous results for $x = 0.33$ (ref. 14).

As already seen in Fig. 3, the checkerboard CO does not change below room temperature in $La_{5/3}Sr_{1/3}CoO_4$. This is consistent with the very robust nature of the checkerboard phase, which persists up to ~825 K in half-doped $La_{3/2}Sr_{1/2}CoO_4$ (ref. 16). In $La_{3/2}Sr_{1/2}CoO_4$, AFM ordering of spins on the $Co^{2+}$ sites (Fig. 1b) sets in gradually at ~40 K and gives rise to strong diffraction peaks close to (0.25, 0.25, l) and (0.75, 0.75, l) with l odd[16,20]. We found no evidence for such AFM order associated with the checkerboard CO phase in $La_{5/3}Sr_{1/3}CoO_4$.

## Discussion

Our measurements indicate that the ground state of $La_{5/3}Sr_{1/3}CoO_4$ is phase-separated into two dominant components with different local hole concentrations: (1) spin and charge stripe order ($x \simeq 1/3$) and (2) checkerboard charge order ($x \simeq 1/2$). The typical size of the correlated regions is $\lesssim 1$ nm. At low temperature the different orders appear static on the timescale of neutron diffraction (~$10^{-12}$ s) and probably also magnetic resonance[12,13] (~$10^{-6}$ s), but, given the short correlation lengths, they must in reality be quasi-static, that is, not static but fluctuating at a rate that is slower than the experimental time window. To maintain charge neutrality, there must also be regions with local doping $x < 1/3$; however, these regions must be small, as we could not detect any distinct diffraction signature of them. In particular, we did not observe any magnetic peaks at $\mathbf{Q}_{AF}$ characteristic of the $La_2CoO_4$-type Néel AFM order.

The results obtained here are significant for understanding the ground state and magnetic dynamics of $La_{2-x}Sr_xCoO_4$, and could also have implications for the copper oxide high-temperature superconductors. The magnetic spectrum of $La_{2-x}Sr_xCoO_4$ for $x = 0.25$–$0.4$ is notable in that its overall structure, that is, the distribution of intensity as a function of energy and in-plane wavevector observed in neutron-scattering measurements, has an hourglass shape[1–3]. The spectrum emerges from the four incommensurate magnetic diffraction peaks that surround $\mathbf{Q}_{AF}$. With increasing energy, the four peaks first disperse inwards until they merge at $\mathbf{Q}_{AF}$ and then disperse outwards again. Above the waist of the hourglass the spectrum also has a fourfold pattern, but now rotated by 45° with respect to the pattern below the waist.

The discovery of quasi-static charge stripes consolidates the stripe scenario proposed to explain the hourglass spectrum in $La_{5/3}Sr_{1/3}CoO_4$ (ref. 1). An hourglass spectrum arises naturally from a stripe-ordered ground state in which magnetic correlations are stronger along the stripes than across them[21,22]. In such a case, the lower and upper parts of the spectrum are from the inter- and intrastripe parts of the magnon dispersion, respectively, and the waist is a saddle point formed from the maximum of the interstripe dispersion and the minimum of the intrastripe dispersion. The 45° rotation of the intensity maxima above the waist relative to the fourfold pattern below the waist is explained by the superposition of quasi-one-dimensional dispersion surfaces from orthogonal stripe domains[1].

Recently, a stripe-free model of $La_{2-x}Sr_xCoO_4$ has been proposed that also has an hourglass spectrum[3,14]. The model is based on a phase-separated ground state comprising nanoscopic $x = 0.5$ regions with checkerboard CO coexisting with undoped ($x = 0$) regions with Néel AFM order[3]. In this picture, the part of the hourglass spectrum below the waist derives from magnetic correlations associated with the checkerboard CO regions, while the part above waist comes from Néel AFM correlations in the the undoped regions.

Let us consider the stripe-free model in the light of the present results. First, in our sample of $La_{5/3}Sr_{1/3}CoO_4$ we found $\epsilon_{so} \simeq \epsilon_{co} = 0.36 \pm 0.01$, which is significantly different from the value $\epsilon_{so} = \epsilon_{co} = 0.5$ characteristic of the AFM order associated with the ideal checkerboard CO (Fig. 1b). In our view, this is clear evidence that the strong magnetic peaks from which the low-energy part of the hourglass spectrum emerges are associated with the stripe component of the ground state and not the checkerboard regions. Second, this evidence is reinforced by the fact that the magnetic diffraction peaks in $La_{5/3}Sr_{1/3}CoO_4$ persist up to ~100 K (Fig. 5 and Supplementary Fig. 4), well above the corresponding magnetic ordering temperature $\simeq 40$ K of the checkerboard CO phase observed of $La_{3/2}Sr_{1/2}CoO_4$ (refs 15, 20). Owing to disorder effects, the magnetic correlations are expected to be weaker in nano-sized patches of checkerboard CO than in bulk $La_{3/2}Sr_{1/2}CoO_4$, and the magnetic dynamics of these spins could be sufficiently slow and diffusive so that they do not influence the observed hourglass spectrum, consistent with the idea that the observed spectrum is dominated by that from the stripe phase. Third, the absence of $La_2CoO_4$-type AFM order means there can be no gap in the spin fluctuation spectrum associated with spin correlations in any undoped regions, since a gap requires broken rotational symmetry. This is inconsistent with the stripe-free model, which requires the fully gapped $La_2CoO_4$-type magnon spectrum in order to produce the waist and upper part of the hourglass. In addition, the magnon spectrum of Néel-ordered $La_2CoO_4$ is isotropic until the spectrum approaches the zone boundary at high energies[23], and therefore cannot reproduce the observed tetragonal anisotropy and 45° rotation of the intensity maxima relative to the fourfold pattern below the waist of the hourglass[1,2].

Although the ground state of $La_{5/3}Sr_{1/3}CoO_4$ is more complex than originally assumed, we conclude that the weight of evidence favours the stripe explanation for the hourglass spectrum. Many hole-doped copper oxide superconductors exhibit an hourglass spectrum with the same general structure as that of $La_{5/3}Sr_{1/3}CoO_4$ (ref. 24), and some underdoped cuprates also have similar (quasi-)static spin and charge stripe order[4,25]. The magnetic spectrum is important for understanding the physics of cuprates because it offers a window on the nature of the correlated magnetic ground state that supports superconductivity. The present results provide an experimental basis for theories that assume a ground state with static or slowly fluctuating stripes in order to explain the hourglass spectrum in cuprates.

## Methods

**Experimental details.** The single-crystal sample of $La_{5/3}Sr_{1/3}CoO_4$ was grown by the floating-zone method in an image furnace and is part of the larger of the two crystals measured previously[1]. Polarized neutron diffraction measurements were performed with the longitudinal polarization set-up of the IN20 triple-axis spectrometer at the Institut Laue—Langevin. Approximately 5 g of the crystal was mounted in a helium cryostat and aligned with $(h, h, l)$ as the horizontal scattering plane. Incident and scattered neutrons of energy 14.7 meV were selected with a Heusler monochromator and a Heusler analyser. Two pyrolytic graphite filters were placed before the sample to suppress second- and third-order harmonic contamination in the incident beam. A magnetic guide field at the sample position ensured that the direction of the neutron spin polarization (**P**) was aligned with the neutron-scattering vector (**Q**), and a spin flipper was placed after the sample. Neutrons whose spin direction reversed (spin-flip scattering) or did not reverse (non-spin-flip (NSF) scattering) on scattering were recorded separately. Measurements were recorded for a 80 s per point per polarization channel, but scans shown in Figs 3 and 4 were repeated several times and averaged. Error bars on the data in Figs 2–5 are s.d.'s obtained from neutron counts.

**Corrections for non-ideal neutron polarization.** In principle, polarization analysis can be used to separate magnetic scattering of electronic origin from structural scattering because when the neutron polarization **P** is parallel to the

scattering vector $\mathbf{Q}$, the spin of the neutron is always flipped in a magnetic interaction and is always unchanged in a coherent nuclear interaction[26]. In reality, inaccuracies due to imperfect neutron polarization and flipping efficiency cause some leakage of the magnetic scattering into the NSF channel and *vice versa*. The quality of the polarization set-up is represented by the flipping ratio $R$, which in our experiment was measured on a nuclear Bragg peak: $R = \left(I_{Bragg}^{NSF} - B^{NSF}\right) / \left(I_{Bragg}^{SF} - B^{SF}\right)$. Here $I^{NSF}$ and $I^{SF}$ are the raw counts in the NSF and spin-flip channels, and $B^{NSF}$ and $B^{SF}$ are corresponding backgrounds (incoherent scattering and instrumental backgrounds). The observed value was $R = 25.7$, which corresponds to 92.5% polarization.

Supplementary Fig. 1 provides an illustration of the separation of magnetic and nuclear scattering before any corrections were made. The plot shows the raw intensities $I^{NSF}$ and $I^{SF}$ measured along the line $(h, h, 3)$ in reciprocal space. The structural diffuse scattering from charge order and the LTO distortion are very small in this particular scan, and the near absence of any structure in $I^{NSF}$ shows that there is very little leakage of the strong magnetic peaks into the NSF channel. The peaks fitted to $I^{NSF}$ and $I^{SF}$ are consistent with $R = 25.7$, which demonstrates that the good polarization observed in Bragg diffraction also extends to diffuse scattering.

As there inevitably is a small leakage between channels, which is almost entirely due to beam polarization, we applied standard corrections to the data given by

$$\begin{pmatrix} N \\ M \end{pmatrix}_{corr} = \frac{1}{R-1} \begin{pmatrix} R & -1 \\ -1 & R \end{pmatrix} \begin{pmatrix} I^{NSF} - B^{NSF} \\ I^{SF} - B^{SF} \end{pmatrix}_{meas},$$

where $N$ and $M$ are the corrected nuclear and magnetic intensities, respectively. $M$ contains the total intensity from magnetic components perpendicular to $\mathbf{Q}$.

Supplementary Fig. 2 illustrates the effect these corrections have on the structural diffuse scattering studied in this work. The upper panels (a, d and g) show uncorrected data. The middle panels (b, e and h) show data corrected with $R = 25.7$, and the lower panels (c, f, i) are corrected with $R = 15$, a value well below that observed in our experiment. In each case, data recorded at 2 and 300 K are shown in the first two columns, and the difference between 2 and 300 K is plotted in the third column for the NSF/$N$ channel. The effect of the polarization corrections is seen to be negligible. This is important because at low temperature the structural diffuse scattering peaks due to the stripe charge order are almost coincident with the magnetic diffraction peaks. The results in Supplementary Fig. 2 demonstrate that the corrections are not very sensitive to the value of $R$, and so one can be confident that the stripe CO signal obtained in this work is not the result of feed-through from the magnetic scattering channel.

**Data availability**. The neutron diffraction data that support the findings of this study are available from the Institut Laue—Langevin with the identifiers (https://dx.doi.org/10.5291/ILL-DATA.5-53-247 and https://dx.doi.org/10.5291/ILL-DATA.DIR-132)[27,28].

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

## Acknowledgements

This work was supported by a European Research Council CONQUEST grant and by the Engineering and Physical Sciences Research Council of the United Kingdom (grant no. EP/J012912/1).

## Author contributions

D.P. prepared and characterized the single-crystal samples. A.T.B., P.B., P.G.F. and M.E. performed the neutron-scattering experiments. P.B. performed the data analysis, and P.B. and A.T.B. wrote the manuscript with input from the other co-authors.

## Additional information

**Competing financial interests:** The authors declare no competing financial interests.

