## [Peer Review File · Nature Communications]

Reviewer #1 (Remarks to the Author):

A. Summary of the key results

The authors have shown that static stripe-like charge and spin order exist in $\text{La}_{5/3}\text{Sr}_{1/3}\text{CoO}_4$, resolving a controversy about the origin of an hourglass-shaped spectrum in the magnetic excitations.

B. Originality and interest: if not novel, please give references

The interplay between stripe-like charge order and superconductivity is of key importance for understanding the origin of superconductivity in the cuprate superconductors, which have the highest transition temperatures of any material at ambient pressure.

The authors have studied a closely related material, $\text{La}_{5/3}\text{Sr}_{1/3}\text{CoO}_4$. Like the cuprate superconductors, LSCO has an hourglass-shaped magnetic excitation spectrum. Much work in the cuprates has gone into understanding the connection between superconductivity and this hourglass spectrum. The authors' work has revealed that the source of the spectrum in LSCO is static charge and spin stripes. This represents a significant advance in our understanding of what may drive superconductivity in cuprates.

The results are original, important, and of broad interest.

C. Data & methodology: validity of approach, quality of data, quality of presentation

The data and methodology are sound, and the presentation is clear.

D. Appropriate use of statistics and treatment of uncertainties

The use of statistics and treatment of uncertainties is appropriate.

E. Conclusions: robustness, validity, reliability

The data reported lead to a solid conclusion that static charge and spin stripes are present in the material. The authors make solid arguments that these stripes are the origin of the hourglass spectrum, consistent with findings in other classes of materials.

F. Suggested improvements: experiments, data for possible revision

The authors have done a thorough job.

G. References: appropriate credit to previous work?

Appropriate references are cited.

H. Clarity and context: lucidity of abstract/summary, appropriateness of abstract, introduction and conclusions

The paper is well-written.

Reviewer #2 (Remarks to the Author):

In this manuscript authors used polarized and unpolarized neutron diffraction to study the spin order (SO) and charge order (CO) in $\text{La}_{5/3}\text{Sr}_{1/3}\text{CoO}_4$. The manuscript reports a discovery of the short-range charge modulation in $\text{La}_{5/3}\text{Sr}_{1/3}\text{CoO}_4$, whose wave vector is roughly twice the wave vector of magnetic structure, matching the popular "stripe" picture. As it turns out, observing this charge order requires clear and profound understanding of the underlying physics, which present authors have demonstrated.

It is for this reason that CO in $\text{La}_{5/3}\text{Sr}_{1/3}\text{CoO}_4$ has not been observed previously in Ref. 14, which lead authors of that paper to draw questionable conclusions based upon an incorrect assumption of its absence.

The manuscript is written very clearly. It presents high quality data, which are competently analyzed, leading to interesting and important results. These results are new and original, and will be of very significant interest to a broad community of researchers in the field of strongly correlated electron systems. The findings of the manuscript are of immediate relevance for a number of recent publications in *Nature* and *Nature Communications*, and therefore publication of this manuscript in *Nature Communications* would be very appropriate. I recommend that the manuscript is accepted for publication after authors make some minor revisions, which I outline below.

1. While I agree with the present authors that interpretations of Ref. 14, which are based upon an incorrect assumption of an absent CO are wrong, I still am not entirely convinced that their observations uniquely indicate the presence of charge stripes. Indeed, it has been argued by A. Savici, et. al., *Phys. Rev. B* 75, 184443, in the context of the slightly incommensurate magnetic order in $\text{La}_{3/2}\text{Sr}_{1/2}\text{CoO}_4$ that presence of one-dimensional stripe-like patterns in real space must translate into a corresponding rotational-symmetry-broken patterns in the reciprocal space. This is different from the type of result presented by a simulation that authors show in Fig. 4a. An alternative to the stripe-type real space picture is that of a mixture of short-range superlattice patches, antiferromagnetically ordered, averaging to $x=1/3$ (to preserve the charge neutrality), such as was proposed by Sakiyama, et. al., *Phys. Rev. Phys. Rev. B* 78, 180406(R) for the case of high hole doping, $x>0.5$ (which is equivalent to electron doping) in $\text{Pr}_{2-x}\text{Ca}_x\text{CoO}_4$. In fact, it appears to me that making a distinction between the two possibilities based upon the results presented in the present manuscript is not possible, and therefore I recommend that authors discuss this alternative.

2. In discussing the models of combined spin and charge order in the last section of the Supplementary Information, authors make their conclusions based upon an observation that "the CO diffuse scattering has maxima close to odd l and minima at even l ". I note, however, that the l -dependence of the CO scattering in Fig. 2d shows extremely broad peaks, whose width indicates that the correlation length is much shorter than the distance between the two layers that authors consider in their models. Hence, such scattering must be insensitive to the CO layer-to-layer stacking. In fact, the observed l -dependence in Fig. 2d is in very close agreement with that governed by the structural distortion of a single CoO_6 octahedron, the model of which was introduced in Ref. 15 of the manuscript. The pattern observed by the present authors seems to be quantitatively very similar to that in the top

right panel of Fig. 1 of Ref. 17, where the correlation length along the c-axis is less than 0.2 lattice repeats, and therefore is insensitive to the type of the CO stacking. Given the insensitivity of the CO scattering to stacking and no evidence for magnetic peaks at even l in a $(1/3, 1/3, l)$ scan, I would suggest that the best stacking choice is model which only gives peaks at odd l , that is model 4.

Reviewer #3 (Remarks to the Author):

The charge stripe instability was first discovered in high- T_c superconductor $\text{La}_{2-x}\text{Sr}_x\text{CuO}_4$ with the single layered Ruddlesden-Popper structure. Its role in the Cooper pairing mechanism remains one of the most important unsettled issues in modern condensed matter physics. The well-known hour-glass shaped spin excitation in hole-doped cuprates hosts the spin resonance mode at its Saddle point, which rises at the commensurate magnetic wave vector positions below the superconducting transition temperatures. The discoveries of similar stripe phase and hour-glass like spin fluctuation in isostructural nickelates suggested the universality of the stripe order in such structural motif. However the absence of such charge stripes in single layered cobaltites challenges a unified picture. The authors of present manuscript have provided the direct observation of such charge stripes in $\text{La}_{1/3}\text{Sr}_{1/3}\text{CoO}_4$. The experimental methods and the data analysis presented here are scientifically sound. The obstacles that challenged the previous attempts were properly addressed. The potential source of false signals, such as the leakage from the spin flip channel due to the imperfection of polarization, was correctly ruled out. So their results are convincing and I agree with the authors that the charge stripes, similar to those seen in cuprates and nickelates, indeed exist in $\text{La}_{1/3}\text{Sr}_{1/3}\text{CoO}_4$. Based on this newly found charge stripe the authors offer an appealing alternative explanation for the hour-glass spectra to the one suggested by authors in Refs. 3 and 14. Whether or not this is correct, the discovery of the charge stripes itself carries enough weight for its publication on Nature Communications. The clarity in data presentation and writing in the manuscript has reached such a level that I don't see the need for any major changes. I only suggest the authors add something short in the supplementary information to address the half lambda or higher order harmonics and how they should not be a concern for the authenticity of the weak signal.

We are grateful for the helpful comments and suggestions from all the referees. We thank the referees for taking time to read through and consider our work and are delighted that all the referees found the work acceptable for the Nature Communications journal. We have modified our manuscript to address the questions raised by Referees 2 and 3.

Reviewer# 2 (Response to reviewer):

1. “While I agree with the present authors that interpretations of Ref. 14, which are based upon an incorrect assumption of an absent CO are wrong, I still am not entirely convinced that their observations uniquely indicate the presence of charge stripes. ... An alternative to the stripe type real space picture is that of a mixture of short-range superlattice patches, antiferromagnetically ordered, averaging to $x=1/3$ (to preserve the charge neutrality), such as was proposed by Sakiyama, et. al., Phys. Rev. Phys. Rev. B 78, 180406(R) for the case of high hole doping, $x>0.5$ (which is equivalent to electron doping) in $\text{Pr}_{2-x}\text{Ca}_x\text{CoO}_4$. In fact, it appears to me that making a distinction between the two possibilities based upon the results presented in the present manuscript is not possible, and therefore I recommend that authors discuss this alternative.”

The referee raises a point that an alternative stripe arrangement is possible where scattering at off-commensurate positions appears from different types of discommensuration related to the nature of stacking faults. The distinction between this and our interpretation in terms of charge stripes manifests itself as a subtle breaking of the four-fold symmetry of the diffuse scattering around the incommensurate magnetic and charge peaks. In fact, we have already observed such a breaking of four-fold symmetry around the magnetic peaks (reported in Ref. 1), but much more complete intensity maps of the CO diffuse scattering would be needed to see a similar effect in the CO peaks. In response to this good point we have added the following on page 4 of the manuscript:

“In Fig. 4a the charge peaks are depicted as isotropic (i.e. circular), but diffuse scattering peaks from stripe order will in general display 2-fold symmetry as a result of different correlation lengths parallel and perpendicular to the stripes. \cite{Savici} The SO peaks display this expected 2-fold anisotropy as reported previously, [1] but more complete data would be needed to resolve any such anisotropy in the correlation lengths for the charge order.”

The referee mentions the possibility of a mixture of commensurate superlattice phases with $x = 1/4, 1/3, 1/2, \dots$. Our findings are indeed consistent with a nanoscale coexistence of $x = 1/3$ and $x = 1/2$, but these order at different temperatures (Fig. 5) and produce peaks at distinct wave vectors. It is possible, however, that the small incommensurability we observe in the stripe CO peak position is due to the presence of defects in an otherwise ideal period-3 stripe phase. To include this possibility we have modified the sentence on page 4:

“...suggests either that the hole doping level of our crystal could be slightly in excess of $1/3$, or that the presence of defects in an otherwise ideal period-3 stripe pattern causes an effective small incommensurability. \cite{Savici}

2. "... Given the insensitivity of the CO scattering to stacking and no evidence for magnetic peaks at even l in a $(1/3, 1/3, l)$ scan, I would suggest that the best stacking choice is model which only gives peaks at odd l , that is model 4."

We agree with the referee that in our interpretation of the out-of-plane stacking we are making the assumption that the charge order is correlated along the c -axis. An alternative view suggested by the referee, which is also valid, is that the perceived charge-ordered peaks originate from the CoO6 octahedra, analogously to what was found in Ref. 15. In such case, either model 1 or model 4 could account for the results. Further to this, it is possible to distinguish between the two different scenarios by performing a scan at $(2/3, 1/3, l)$ which could be examined in future experiments. We have modified the last paragraph of the Supplementary Information as follows:

"However, this conclusion should be considered tentative as the peaks are very broad in l and it is possible that the observed l -dependence comes entirely from the form factor of the CoO6 distortion pattern, and not from any inter-layer correlations. A direct way to check this would be to measure the l -dependence along $(2/3, 1/3, l)$. If inter-layer correlations are significant then the non-magnetic scattering along $(2/3, 1/3, l)$ should be different from that along $(1/3, 1/3, l)$ in accord with the structure factors given in Table I. Temperature difference data to fully isolate the l -dependence of the stripe CO scattering from the tails of the checkerboard CO peaks would also be valuable."

Reviewer# 3 (Response to reviewer):

1. "I only suggest the authors add something short in the supplementary information to address the half lambda or higher order harmonics and how they should not be a concern for the authenticity of the weak signal."

Our central interpretation of the diffraction data in terms of charge stripes is obtained from analysis of difference scans in which measurements at 300 K are subtracted from 2 K data. This means that the neutron harmonics, which are temperature independent, will cancel and not influence our results. To highlight that we have considered this contamination to our signal, we have added a label and arrows in Fig. S1 to show the half-lambda contribution and have added the following text at the end of the caption of Fig. S1:

"... are temperature-independent and so cancel out when scans at different temperatures are subtracted, as was done to reveal the stripe CO scattering." We believe that our revisions have improved the paper and trust that we have answered the comments of the referees satisfactorily. We hope that the manuscript is now suitable for publication.

We believe that our revisions have improved the paper and trust that we have answered the comments of the referees satisfactorily. We hope that the manuscript is now suitable for publication.

Reviewer #2 (Remarks to the Author):

The manuscript reports polarized and unpolarized neutron diffraction to study the spin order (SO) and charge order (CO) in $\text{La}_{5/3}\text{Sr}_{1/3}\text{CoO}_4$. In this work authors discovered the short-range charge modulation in $\text{La}_{5/3}\text{Sr}_{1/3}\text{CoO}_4$, whose wave vector is roughly twice the wave vector of magnetic structure, matching the popular "stripe" picture. As it turns out, observing this charge order requires clear and profound understanding of the underlying physics, which present authors have demonstrated. It is for this reason that CO in $\text{La}_{5/3}\text{Sr}_{1/3}\text{CoO}_4$ has not been observed previously in Ref. 14, which lead authors of that paper to draw questionable conclusions based upon an incorrect assumption of its absence.

As I have noticed in my original review, the manuscript presents high quality data, which are competently analyzed, leading to interesting and important results. These results are new and original, and will be of very significant interest to a broad community of researchers in the field of strongly correlated electron systems. The findings of the manuscript are of immediate relevance for a number of recent publications in Nature and Nature Communications, and therefore publication of this manuscript in Nature Communications would be very appropriate. I recommended that the manuscript is accepted for publication after authors make some minor revisions.

In the revised version authors have thoroughly addressed my comments and suggestions, which has further improved the manuscript. I therefore recommend that the manuscript is accepted for publication in Nature Communications.

Reviewer #3 (Remarks to the Author):

I believe the authors have properly addressed my concern as well as the other referee's.